# Effects of Eight-Week Circuit Training with Core Exercises on Performance in Adult Male Soccer Players

Guido Belli [1], Sofia Marini [1] 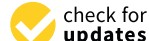, Mario Mauro [1,*] , Pasqualino Maietta Latessa [1] and Stefania Toselli [2]

1   Department of Life Quality Studies, University of Bologna, 47921 Rimini, Italy
2   Department of Biomedical and Neuromotor Sciences, University of Bologna, 40126 Bologna, Italy
*   Correspondence: mario.mauro4@unibo.it

**Abstract:** Core exercises have been widely promoted in the last 25 years. However, the scientific debate about its efficacy for improving individual and team sports performance is still open. Thus, the present study aims to investigate the effects of circuit training with a core exercise program on physical performance in competitive amateur soccer players. The training was conducted during the off-season period, two times per week for 8 weeks. Pre- and post-evaluations were conducted using the following tests: Y-Balance Test (YB), standing long jump (SLJ), medicine ball chest press (MBC), curl-up (CU), and Illinois Agility Test (IAT). A total of 19 adults were divided into an experimental group (EG, n = 11, age 22 years, weight 71.2 ± 4.8 kg, height 174 ± 5.8 cm) and a control group (CG, n = 8, age 22 years, weight 73.2 ± 4.1 Kg, height 176 ± 6.3 cm). The EG showed significant improvements in lower and upper body strength, core endurance and balance, whereas the CG did not report significant changes in the pre- and post-test comparison. Despite study limitations, our positive results show that circuit training with core exercises appears to be a good strategy for performance improvement in adult soccer players.

**Keywords:** core exercise; soccer; circuit training; strength; balance

## 1. Introduction

Circuit training (CT) is a popular methodology in fitness and wellness programs, as well as in sports, because its modulation induces physiological benefits, such as strength, power, and cardio-vascular–respiratory adaptations [1–5]. A circuit includes a variable number of exercises that come in succession with a specific time related to the adaptation to the relevant exercises [6]. Each exercise should be general and/or sport-related and can involve the whole body or just a specific body compartment [6,7].

Core exercises have been widely promoted in the last 25 years. The role of the core musculature and several training methods have been investigated in sport, fitness, and rehabilitation to understand how trunk conditioning could affect performance and health [8–15]. The core region, also identified as the lumbo-pelvic-hip complex and scapular stabilizing system, is the central part of the body that connects the trunk with upper and lower limbs and represents the center of myofascial kinetic chains [9,10,13,14]. Several studies showed that core exercises improve stability and neuromuscular control between the spine and pelvis, and increase endurance, strength, and power in trunk muscles [16–18]. Additionally, they could facilitate the force transfer between the upper and lower body, increase static and dynamic balance, and improve the execution of specific sports skills [10,13,14,17]. Consequently, a proper level of core stability and strength could improve sports performance and physical fitness, prevent several musculoskeletal injuries, and optimize training adaptations in athletes [12,13,18–20].

Despite the large diffusion and benefits of core exercises in training programs for individual and team sports, the scientific debate about its efficacy is still open [8,18,20,21]. Some authors highlighted how core exercises induced positive effects on neuromuscular

control, postural stability, trunk muscle strength, and endurance, but it remains unclear whether these improvements could be transferred into sport-specific performance [21]. Differently, a recent review showed that the core program could improve specific skills in team and individual sports if practiced at least twice a week for a month [20]. In addition, a meta-analysis exhibited several benefits for general physical fitness and sport-specific performance after more than 18 short core training sessions [18].

As regards team sports, soccer has been classified as an intermittent sport that requests many types of physical abilities, such as strength, power, endurance, balance, and several coordinative and technical–tactical skills [22,23]. The CT and core exercises performed by soccer players showed several positive effects on static and dynamic balance [24], lower limbs strength [25–28], speed and agility [17,24,25,27–32], muscularity in the trunk and cross-sectional area [17,25–27,33], and flexibility [28]. However, some authors reported that both static and dynamic core exercises enhanced trunk stability without positive effects on speed, agility and quickness, and lower limb strength and power [34].

Although the current literature explained conflicting results of the addition of core exercises to general and specific soccer training in all levels of players, we hypothesize that proper CT including core exercises could report positive effects on balance, strength, and power. So, this study aims to analyze the effects of a specific CT protocol, performed during the off-season time, on trunk, upper and lower body strength, dynamic balance, and speed in competitive amateur soccer players.

## 2. Materials and Methods

### 2.1. Subjects and Study Design

This is a longitudinal study design of eight weeks with two evaluation times (pre and post), from May up to July (off-season period). The study was conducted at the Sports Science Institute of Bologna, Italy. The participant's enrolment was conducted within the soccer team Madonnina Calcio (Modena, Italy) at the end of the regular season (Emilia Romagna regional league, Italy). The Madonnina Calcio team comprised 23 amateur soccer players. Players were defined as an amateur if they trained less than three times per week and played a maximum of one match per week. Additionally, their performances were not compensated. The eligible criteria were: (a) no history of musculoskeletal, neurological, or other orthopaedic disorders in the last 6 months, (b) age range 18–30 years, and (c) no previous experience with core training exercises. In the beginning, 23 players were considered to participate in the study. Of these, three players were injured, whereas one player could not guarantee their participation due to the holiday. Nineteen participants volunteered for the study and completed all the evaluations (Figure 1). No randomization was possible because many participants could not guarantee they could perform at least 12 CT training sessions. Participants were allocated to one of two groups: the experimental group (EG, n = 11, age 22 years, weight 71.2 ± 4.8 kg, height 174 ± 5.8 cm) and the control group (CG, CG, n = 8, age 22 years, weight 73.2 ± 4.1 Kg, height 176 ± 6.3 cm). No nutritional information was collected. Written informed consent was provided before the beginning of the study. The research was approved by the Bioethics Committee of the University of Bologna (Approval code: 25027).

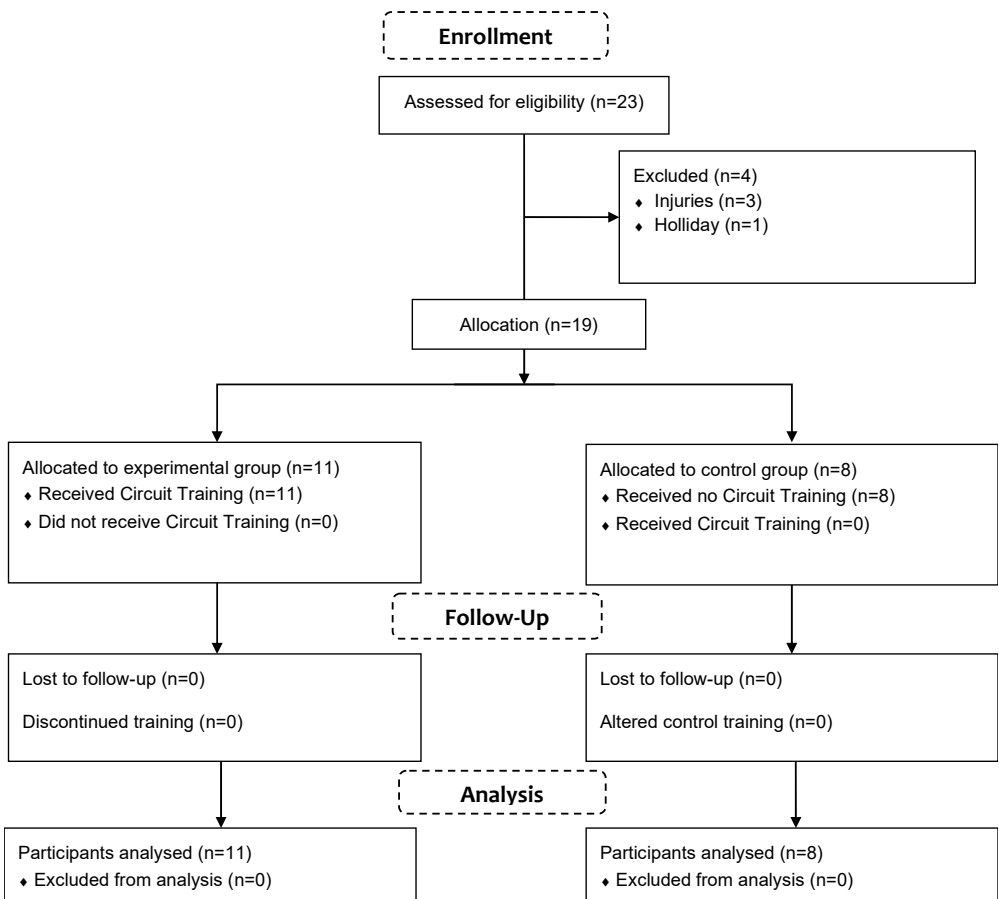

**Figure 1.** Participants' flowchart.

## 2.2. Training in the EG and CG

The EG was submitted to an eight-week core and functional training protocol [27,31]. Training frequency was two sessions per week and the global number of sessions was sixteen, according to previous research that demonstrated benefits in neuromuscular control and performance measures during a similar conditioning period [35]. The time of rest between each training session was 72 h to optimize the physiological recovery (training days were Monday or Tuesday–Thursday or Friday).

Figure 2 shows the training protocol progression. It was focused on seven exercises performed in the following way:

| Exercise | Core Stability | | Core Endurance | | Core Strength | | Strength + Endurance | |
|---|---|---|---|---|---|---|---|---|
| Plank Bridge Crunch Side Plank | Isometric | Instability | Volume | Volume Instability | Intensity | Intensity Instability | Volume Intensity Instability | Rotary Stability |
| Push Pull Press Squat Deadlift Lunge | Fundamental | Unilateral | Volume | Volume Unilateral | Weight | Weight Instability | Volume Plyometric Instability | Rotary Stability |
| | Week 1 | Week 2 | | | → | | Week 7 | Week 8 |

**Figure 2.** Training protocol progression.

Session 1: four core training and three upper body strength exercises
Session 2: four core training and three lower body strength exercises

Core training exercises protocol (Figures 2 and 3) was gradually focused on trunk stability, endurance, and strength, while load intensity and volume were weekly increased following previous guidelines [14–17,36]. The four exercises performed were the plank, crunch, supine bridge, and side bridge [37].

| | | WEEK 1 | | WEEK 2 | |
|---|---|---|---|---|---|
| | | ① CORE STABILITY UB (3 CT) | ② CORE STABILITY LB (3 CT) | ③ CORE STABILITY UP (3 CT) | ④ CORE STABILITY LB (3 CT) |
| E | 1 | PLANK (FOREARMS) | PLANK (FOREARMS) | PLANK WTH FITBALL | PLANK WTH FITBALL |
| X | 2 | PRESS WITH WEIGHT PLATE | SQUAT WITH WEIGHT PLATE | UNILATERAL PRESS WITH WEIGHT PLATE | SIDE SQUAT WITH WEIGHT PLATE |
| E | 3 | SUPINE BRIDGE | SUPINE BRIDGE | 1 LEG SUPINE BRIDGE | 1 LEG SUPINE BRIDGE |
| R | 4 | PUSH UP | DEADLIFT WITH ELASTIC BAND | 1 LEG PUSH UP | DEADLIFT WITH WEIGHT PLATE |
| C | 5 | CRUNCH | CRUNCH | CRUNCH WITH FITBALL | CRUNCH WITH FITBALL |
| I | 6 | TRX BACK ROW | BACK LUNGE | TRX SINGLE ARM BACK ROW | SIDE LUNGE |
| S E | 7 | SIDE PLANK (FOREARM) | SIDE PLANK (FOREARM) | 1 LEG SIDE PLANK (FOREARM) | 1 LEG SIDE PLANK (FOREARM) |
| | | WEEK 3 | | WEEK 4 | |
| | | ⑤ CORE ENDURANCE UB (4 CT) | ⑥ CORE ENDURANCE LB (4 CT) | ⑦ CORE ENDURANCE (4 CT) | ⑧ CORE ENDURANCE (4 CT) |
| E | 1 | PLANK (HANDS) | PLANK (HANDS) | PLANK WITH FITBALL (HANDS) | PLANK WITH FITBALL (HANDS) |
| X | 2 | PRESS WITH SANDBAG | SQUAT WITH SANDBAG | UNILATERAL PRESS WITH SANDBAG | SIDE SQUAT WITH SANDBAG |
| E | 3 | SUPINE BRIDGE (FEET ON MEDBALL) | SUPINE BRIDGE (FEET ON MEDBALL) | 1 LEG SUPINE BRIDGE WITH MEDBALL | 1 LEG SUPINE BRIDGE WITH MEDBALL |
| R | 4 | PUSH UP (HANDS ON MEDBALLS) | DEADLIFT WITH WEIGHT PLATE | PUSH UP (1 HAND ON MEDBALL) | CLEAN WITH MEDBALL |
| C | 5 | SIT UP | SIT UP | SIT UP WITH FITBALL | SIT UP WITH FITBALL |
| I | 6 | TRX PULLDOWN | TRX BACK LUNGE | TRX SINGLE ARM PULLDOWN | TRX SIDE LUNGE |
| S E | 7 | SIDE PLANK (HAND) | SIDE PLANK (HAND) | 1 LEG SIDE PLANK (HAND) | 1 LEG SIDE PLANK (HAND) |
| | | WEEK 5 | | WEEK 6 | |
| | | ⑨ CORE STRENGTH (3 CT) | ⑩ CORE STRENGTH (3 CT) | ⑪ CORE STRENGTH (3 CT) | ⑫ CORE STRENGTH (3 CT) |
| E | 1 | MOUNTAIN CLIMBER | MOUNTAIN CLIMBER | MOUNTAIN CLIMBER WITH FITBALL | MOUNTAIN CLIMBER WITH FITBALL |
| X | 2 | PRESS WITH BARBELLS | OVERHEAD SQUAT WITH WEIGHT PLATE | UNILATERAL FORWARD PRESS WITH BARBELL | SINGLE LEG OVERHEAD SQUAT WITH WEIGHT PLATE |
| E | 3 | REVERSE MOUNTAIN CLIMBER | REVERSE MOUNTAIN CLIMBER | KNEE TUCK WITH FITBALL | KNEE TUCK WITH FITBALL |
| R | 4 | SPIDERMAN PUSH UP | DEADLIFT WITH BARBELLS | SPIDERMAN PUSH UP WITH MEDBALLS | CLEAN WITH BARBELL |
| C | 5 | SIT UP WITH WEIGHT PLATE | SIT UP WITH WEIGHT PLATE | SIT UP WITH MEDBALL | SIT UP WITH MEDBALL |
| I | 6 | TRX BACK ROW (TRUNK EXTENSION) | BACK LUNGE WITH WEIGHT PLATE | TRX BACK ROW (FULL TRUNK EXTENSION) | TRX BACK LUNGE WITH WEIGHT PLATE |
| S E | 7 | RUSSIAN TWIST WITH MEDBALL | RUSSIAN TWIST WITH MEDBALL | CROSS OVER CRUNCH | CROSS OVER CRUNCH |
| | | WEEK 7 | | WEEK 8 | |
| | | ⑬ CORE COMBO (4 CT) | ⑭ CORE COMBO (4 CT) | ⑮ CORE COMBO (4 CT) | ⑯ CORE COMBO (4 CT) |
| E | 1 | PIKE WITH FITBALL | PIKE WITH FITBALL | 1 LEG PIKE WITH FITBALL | 1 LEG PIKE WITH FITBALL |
| X | 2 | PRESS WITH DUMBELLS | JUMPING BACK LUNGE | ASYMMETRICAL PRESS WITH BARBELL | FRONT LUNGE AND TWIST WITH WEIGHT PLATE |
| E | 3 | OVERHEAD SANDBAG ROTATION | OVERHEAD SANDBAG ROTATION | 1 LEG OVERHEAD SANDBAG ROTATION | 1 LEG OVERHEAD SANDBAG ROTATION |
| R | 4 | PLYOMETRIC PUSH UP | JUMPING SQUAT | PUSH UP AND TRUNK TWIST | SQUAT AND TWIST WITH MEDBALL |
| C | 5 | SIT UP WITH MEDBALL THROW | SIT UP WITH MEDBALL THROW | SIT WITH MEDBALL LATERAL THROW | SIT WITH MEDBALL LATERAL THROW |
| I | 6 | JUMP AND PULL WITH BAR | SWING WITH KETTLEBELL | TRX SINGLE ARM ROW AND TRUNK TWIST | SINGLE CLEAN WITH KETTLEBELL |
| S E | 7 | WOODCHOP WITH MEDBALL | WOODCHOP WITH MEDBALL | WOODCHOP WITH LOOPBAND | WOODCHOP WITH LOOPBAND |

**Figure 3.** Exercise progression over 16 training sessions (eight weeks).

Upper and lower body functional exercises were defined by sport movement patterns [38]. The three upper body functional movements executed were push, pull, and press, while lower body movements were squat, lunge, and deadlift [27,31,35].

All movements were progressed using specific equipment (unstable surfaces, sling tools, sandbags, weight plates, dumbbells, barbells and kettlebells, medicine balls, elastic bands), and training load parameters were increased in a similar way to core training exercises (Figures 2 and 3). During each exercise, execution participants were supposed to reach maximum effort supported by a trained kinesiologist.

Each session was organized using the circuit interval training method with a ratio between work and rest equal to 1:1, of 30 seconds respectively. This ratio was selected to enhance the stimulation of both anaerobic systems (glycolytic and neuromuscular) and to induce the neuromuscular learning effect (improvement in body efficiency and movement economy) [39,40]. Specifically, the 30″ work period aimed to maintain a selected level of mechanical force (static exercise) and/or power (dynamic exercise) for a short interval (critical power), and to increase the anaerobic energy source (W′) [41]. The passive 30″ recovery period aimed to dilate the exhaustion time and induce a higher phosphorylcreatine resynthesis [42]. Static exercises were performed by holding isometric positions for 30″, whereas dynamic movements requested the execution of the maximum number of repetitions with resistance tools for the same time.

Participants completed three (weeks 1, 2, 5, 6) or four circuits (weeks 3, 4, 7, 8) without additional recovery, and the working time was 21 or 28 min, respectively.

Before starting each training session, 10 min of dynamic warm-up with a focus on cardiovascular activation, joint mobility, and flexibility were performed (bodyweight exercises). Once circuit interval training was completed, 10 min of cool down with stretching and myofascial release exercises concluded each session.

All training sessions were led by a kinesiologist with at least 3 years experience in core and functional training. Before starting the CT protocol, each participant was instructed to learn the correct exercise execution and breathing.

Contrary to the EG, the CG performed recreational activities, such as running, biking, and futsal, for the same period. CG participants were not allowed to perform any kind of resistance training exercises or bodyweight exercises.

### 2.3. Motor Tests

Motor tests were implemented at the University sports center. Each evaluation was performed before and after the training period in indoor spaces, where the temperature was 21.5°. All testing was performed in the afternoon between 6 p.m. and 9 p.m. due to participants' availability, similar to training time sessions. Different days were selected to perform strength and balance tests, and 24 to 48 h were given to participants to avoid biased results due to fatigue. According to previous reviews, the following tests were selected [13,18,20,43]: Y-Balance Test (YB), standing long jump (SLJ), medicine ball chest press (MBC), curl-up (CU), Illinois Agility Test (IAT). Before performing each test, participants were instructed by a trained specialist who taught them the accurate and safe procedure. Each participant did a warm-up of 15 min with jogging, dynamic stretching, and many athletic drills, such as jumps, skips, lunges, short-run shuffles, core stimulations, arms swings, and wide push-ups, after which they practiced some trials to get comfortable with the specific test. Following this dynamic warm-up, a period of eight minutes was allotted to the participants. Three trials were completed for each evaluation, separated by 4 min of rest intervals. Only the best results were recorded.

#### 2.3.1. Standing Long Jump Test (SLJ)

Horizontal jump tests measured the explosive strength and power in the lower limb. These aspects are fundamental to the usual movements of soccer players, such as sprinting, jumping, and kicking a ball [22]. The role of the core musculature can optimize the performance during this test [11,13,19].

For the SLJ, all subjects received standardized instructions that allowed them to begin the jump with bent knees and to swing their arms to assist in the jump. A line drawn on a hard surface served as the starting line. The length of the jump was determined using a tape measure, which was affixed to the floor. The distance of the best jump was measured, to the nearest 1 cm, from the line to the point where the heel closest to the starting line landed. If the subject fell backwards, the distance where the body part closest to the starting line touched the ground was measured as the jump's length. The reliability and validity of the SLJ in younger participants were previously reported [44,45].

#### 2.3.2. Medicine Ball Chest Test (MBC)

Medicine ball throw tests have been reported to be a valid and reliable measure of upper body strength and power [11]. Furthermore, static, and dynamic throws are significantly correlated with some performance measures and can reflect the force transfer between the core and limbs [46]. Since soccer players require strength and stability during rotary movements, the MBC was performed dynamically in a standing position [47]. Participants kept a split stance position with the front foot in contact with a starting line on the floor and held a 6 Kg medicine ball in their hands. Then, they threw the medicine ball as far as possible with a chest pass. Participants were allowed to rotate the trunk before the throw without moving their feet. The test was performed with the right and left foot in an anterior position. The throws were marked at the first contact on the ground and the distance from starting line was determined using a tape measure.

#### 2.3.3. Curl-up Test (CU)

Curl-up tests measure the strength and endurance of the core musculature [48] and represent an assessment tool for the Fitnessgram® program [49].

For this test, participants attempted to complete up to 75 curl-ups at a specified pace (1 curl for every 3 s, 20 reps per minute) using a mat with a 12 cm measuring strip. They lay flat on their backs with their knees bent at 90° and feet flat on the floor. Arms were extended and parallel to the trunk with their palms on the mat. The measuring strip was used to help participants know how far to curl-up and was placed under the knees with the fingers touching the nearest edge. Participants slid their fingers from one side of the measuring strip to the other side and then curled back down. CU speed was defined using a metronome settled at 40 bpm. The test ended when participants could not keep the requested speed or feet were moved from the mat. The final score was the number of curl-ups completed.

### 2.3.4. Illinois Agility Test (IAT)

The IAT was administered as previously described [50]. The length of the IAT rectangle was set at 9.144 m and the width at 5 m. The IAT course was marked by cones, with four cones spaced 3.05 m apart in a central position and four corner cones positioned 2.5 m laterally from the central cones. The participant began the test lying prone on the floor behind the starting line with his arms at his side and his head turned to the side or facing forward. On the vocal starting command, the participant ascended to his feet and ran or moved quickly forward to the first tape mark. Participants were required to touch or cross the tape mark with their feet. The participant turned around and moved back to the first central cone. The participant then ran or moved as quickly as possible to the second tape mark on the far line. Again, participants were required to touch or cross the end-line tape marks with their feet. Lastly, the participant turned around and ran or moved as quickly as possible across the finish line. The time taken to complete each trial was recorded in seconds.

### 2.3.5. Y-Balance Test (YB)

The YBT was administered as previously described [51]. The test was assessed for both right and left lower limbs. The starting position saw the participant standing on the central footplate, with the distal aspect of the right foot at the starting line. While maintaining a single leg stance on the right leg, the subject moved the free limb to the anterior, posteromedial, and posterolateral directions with the stance foot by pushing the indicator box as far as possible. To reduce fatigue that could negatively affect the test result, participants altered their right and left lower limbs between the three directions. Attempts were discarded and repeated (a maximum of six trials) if the subject failed to maintain a unilateral stance on the platform, failed to maintain reach foot contact with the reach indicator on the target area while the reach indicator is in motion, used the reach indicator for stance support, or failed to return the reaching foot to the starting position under control. The reach distance was recorded to the nearest 0.5 cm.

### 2.4. Statistical Analysis

Descriptive statistics (mean ± standard deviation, SD) were calculated for each variable. Variable normality was verified with the Shapiro–Wilk test. The paired Student's *t*-test was performed to assess the differences between the groups from the pre- to post-evaluation. The student's *t*-test was performed to assess the between-groups differences. To evaluate the different treatment effects with no bias due to beginning participants' heterogeneity, the difference between the pre-and post-evaluation of each group was calculated, and then these differences were inferred. A statistical type I error (*p*-value, *p*) <0.05 was considered significant. A post hoc analysis was computed to achieve the statistical power for both the matched paired and the two group's *t*-test with G*Power 3.1.9.7 for Windows 10 (Heinrich-Heine-Universitat Düsseldorf, Universitätsstraße 1, 40225 Düsseldorf, Germany): for matched paired comparison in the EG the mean ES = 1.368, $\alpha = 0.05$, n = 11, test two-tailed, 1-$\beta$ = 0.982; for matched paired comparison in the CG the mean ES = 0.35, $\alpha = 0.05$, n = 8, test two-tailed [24,25,27–32,34,52], 1-$\beta$ = 0.16; for two

independent groups comparisons, the mean ES = 0.99, α = 0.05, n = 19, test two-tailed, 1-β = 0.52.

The statistical analysis was performed with STATA® software for Windows 10, version 17 (Publisher: StataCorp. 2021. Stata Statistical Software: Release 17. College Station, TX, USA, StataCorp LP).

## 3. Results

Table 1 shows the means, standard deviations, and differences within and between the groups in pre- and post-evaluation. The EG improved significantly in all tests over time except for the Illinois Agility Test, whereas the CG did not show significant differences among pre- and post-evaluation. Within-group comparisons, greater improvements were found in the right-side (dominant body-side) tests of the experimental group. The differences between each group difference showed a positive trend for the EG, but only three tests exhibited significant differences between groups (medicine ball chest press on the right side and the Y-Balance test). Figure 4 includes five graph bars (A–E), which show the means and mean differences of each test for the two groups, respectively.

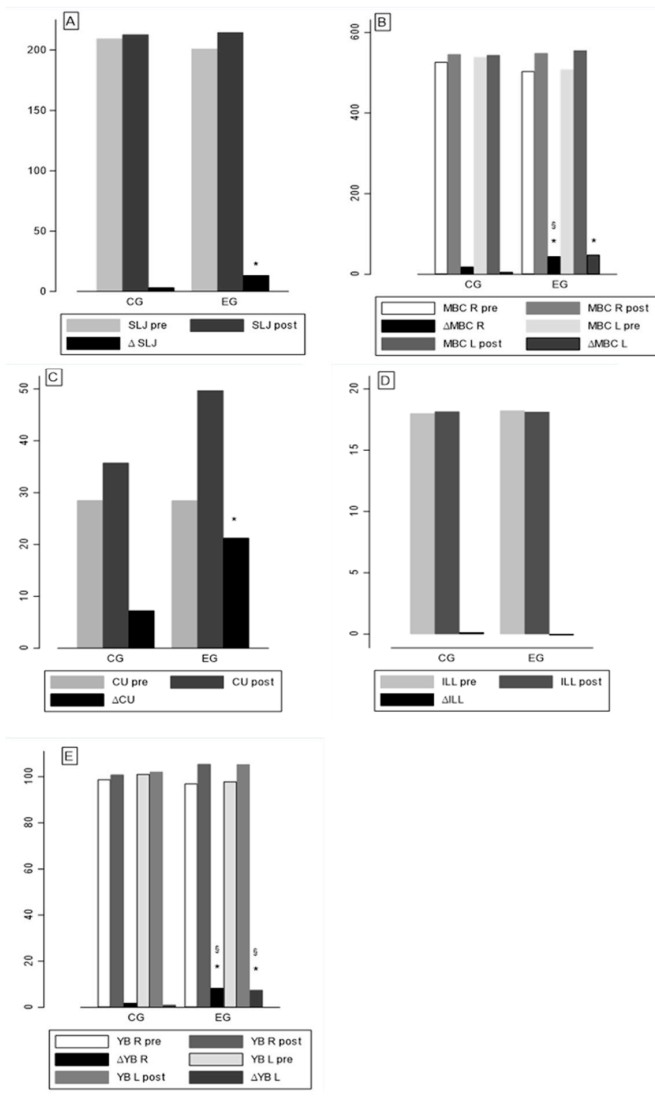

**Figure 4.** Graph bar with pre-, post- and post–pre of standing long jump test (**A**), medicine ball chest test (**B**), curl-up test (**C**), Illinois Agility Test (**D**), and Y-Balance Test (**E**). Note: *, significant difference within group; §, significant difference between groups.

**Table 1.** Summary statistics of motor test results and differences within and between groups.

| | Within Groups | | | | | | | | Between Groups | |
| | EG (*n* = 11) | | | | CG (*n* = 8) | | | | Pre (EG–CG) | Post (ΔEG–ΔCG) |
| Variable | Pre (Mean ± SD) | Post (Mean ± SD) | ΔEG (Mean ± SD) | p. *t* (10) | Pre (Mean ± SD) | Post (Mean ± SD) | ΔCG (Mean ± SD) | p. *t* (7) | *t* (17) | *t* (17) |
|---|---|---|---|---|---|---|---|---|---|---|
| SLJ | 201.09 ± 11.89 | 214.63 ± 11.65 | 13.55 ± 9.33 | 4.814 ‡ | 209.5 ± 14.17 | 212. 87 ± 13.91 | 3.37 ± 12.85 | 0.743 | −1.406 | 2.005 |
| MBCr | 503.64 ± 48.22 | 548.18 ± 41.67 | 44.55 ± 22.52 | 6.559 ‡ | 526.87 ± 40.61 | 545.62 ± 45.15 | 18.75 ± 32.60 | 1.627 | −1.105 | 2.046 * |
| MBCl | 507.27 ± 52.74 | 555.45 ± 54.84 | 48.18 ± 46.87 | 3.409 * | 538.12 ±43.75 | 543.75 ± 56.29 | 5.62 ± 46.40 | 0.343 | −1.348 | 1.962 |
| CU | 28.45 ± 12.19 | 49.73 ± 23.58 | 21.27 ± 17.31 | 4.076 § | 28.5 ± 10.46 | 35.75 ± 10.82 | 7.25 ± 11.77 | 1.742 | −0.009 | 1.976 |
| Ill | 18.22 ± 0.82 | 18.13 ± 0.57 | −0.09 ± 0.77 | 0.382 | 17.99 ± 0.62 | 18.14 ± 0.82 | 0.145 ± 0.61 | 0.667 | 0.635 | 0.709 |
| YBr | 97.05 ± 5.92 | 105.46 ± 5.3 | 8.41 ± 4.44 | 6.285 ‡ | 98.84 ± 6.15 | 100.84 ± 5.35 | 1.99 ± 3.99 | 1.417 | −0.640 | 3.241 § |
| YBl | 97.92 ± 6.46 | 105.41 ± 4.45 | 7.50 ± 4.18 | 5.941 ‡ | 101.11 ± 3.44 | 102.14 ± 3.99 | 1.03 ± 2.04 | 1.429 | 1.267 | 4.016 ‡ |

Note: SLJ, standing long jump test; MBCr, medicine ball chest test right side; MBCl, medicine ball chest test left side; CU, Curl-p test; Ill, Illinois Agility Test; YBr, Y-Balance Test right side; YBl, Y-Balance Test left side; EG, experimental group; CG, control group; *n*, sample size; SD, standard deviation; p. *t*, paired *t*-test; *t*, Student's test; *p*, *p*-value; Δ, difference; *, $p < 0.05$; §, $p < 0.01$; ‡, $p < 0.001$.

## 4. Discussion

The present work aimed to investigate the effects of a circuit training with a core exercise program on physical performance in competitive amateur soccer players. The training was conducted during the off-season period, two times per week for 8 weeks. By our hypothesis, we found significant improvements in lower and upper body strength (SLJ and MDC on both sides, respectively), core endurance (CU), and balance (YBT on both side) in the EG. On the contrary, the CG did not report significant changes in pre–post-test comparisons.

The effects of core exercises on performance and physical fitness in different levels of male soccer players have been previously investigated. Training protocol duration and frequency varied normally from six to twelve weeks and from two to four times per week, respectively [24,25,27–32,34,52]. Recently, a systematic review and meta-analysis evidenced the efficacy of short-time CT programs (<30′) performed twice a week for at least 18 sessions [18]. In the current study, the CT time ranged from 21 to 28 minutes and the total number of sessions was 16, close to the above-mentioned suggestion. Concerning duration, an 8-week integrated training protocol performed two times per week demonstrated an enhancement of neuromuscular control and agility, trunk, and upper body strength when compared to isolated training [35]. This training time could be long enough to provide improvements in neural aspects of strength, coordination, and motor control. Consequently, gains in strength, balance, and agility can be expected. The current literature reports the efficacy of a similar training period in soccer [27,30–32,53,54]. Recently, Mahmoud found significant improvements both in lower (SLJ and vertical jump) and upper (medicine ball throw) arms after a 10-week core exercise program with two sessions per week in younger soccer players (15.40 years) [47]. Additionally, two researchers showed that two different 8-week core exercise programs (static and dynamic) applied for 30 min, two days per week, increased strength levels measured by the SLJ and push-up test in younger soccer players (15 years) [54]. Although the above-mentioned studies investigated younger samples and the upper arm strength was evaluated differently, core exercises induced benefits in strength. In our study, the SLJ and MBC significantly improved in the EG only. The selection of exercises, load progression, and training length could be the main reason for this positive change [31,35]. Furthermore, between-group comparisons highlighted differences in the MBC for the right side only. This result can be related to the physiological discrepancy in the right and left parts of the body and specific movement patterns during throwing [47].

As regards core endurance and strength, this study showed a significant improvement in the experimental group after the 8-week CT with a core exercise program in the curl-up test. Since the focus of the training protocol was mainly on the core musculature, this result was expected. Despite no significant results appearing in the between-groups comparison, the type one error value is near the critical level selected, and a larger sample should exhibit a statistical discrepancy. However, several studies are in accordance with our results and reported improvements after different periods of a core exercise program [17,47,53,54]. Afyon et al. administered a 12-week core exercise plan to 15 younger soccer players (U-16) in addition to regular training. Compared to the control group (soccer training only), the authors reported significant improvements in the plank test and larger benefits in lower body strength (standing long jump and vertical jump), balance, and speed [53]. In addition, Turna et al. showed significant differences in the 30 s abdominal crunch test after six weeks of core training in adolescent soccer players [55]. Prieske et al. compared two different core training programs (stable vs. unstable) in U-17 elite soccer players and found similar improvements in maximal isometric trunk strength for extensor muscles [17]. Since the CU represents an easy and dynamic method to assess core endurance in a different population, this test was preferred to the plank test or the McGill test [48]. Furthermore, the crunch exercise was selected within CT protocols for the muscular activation of the anterior core region [37].

Concerning agility, the EG did not show improvements after the treatment. These results are in line with the study of Server et al. [34], which analyzed the effects of two

different core training protocols (static and dynamic) executed three times per week for an 8-week training period in young male soccer players and did not report significant improvements in sprint and agility. On the contrary, Doganay et al. evidenced benefits in quickness and agility (measured with the Hexagon test) after eight weeks of both core and soccer training performed three times per week [30]. Similarly, Akif showed enhancements in agility (IAT and T-Drill agility tests) with two core exercise sessions per week in amateur players [32]. Although divergent results are presented in the literature, many important elements, such as training protocols (static vs. dynamic exercises, bodyweight vs. strength equipment), motor tests, and season period (in-season vs. off-season), could affect adaptations. The lack of improvements in IAT in the present work could be mainly related to the off-season period: while static and dynamic exercises were added to a sport-specific routine during the regular season [27,30,32], our research focused on core and functional exercises alone. The absence of specific agility and quickness training in our protocol is probably the reason for this lack. Consequently, the specificity of core exercise programs, in addition to soccer training, has been related to different benefits [31].

The last evaluation of this study focused on whether core exercise protocol could improve single-leg dynamic balance measured by the Y-Balance Test. Our results showed that eight weeks of circuit training with core exercises induced benefits in both the right and left sides of the body. The YBT requires lower limb strength, neuromuscular control, flexibility, and an adequate level of core stability [51,52]. Consequently, this assessment tool has been widely reported as an indirect measure of core efficiency and its role in physical fitness is well documented [11,12]. Imai et al. compared the effects of two different trunk training programs on balance and other performance parameters [24]. The authors found significant improvements in the posteromedial and postero-lateral direction of the YBT after 12 weeks of core training exercises (front plank, back bridge, side bridge, and quadrupeds' arm–leg extension) when compared to traditional trunk exercises (sit up and back extension). Additionally, some researchers showed that eight weeks of core training assessed three times per week (in season) improved the balance control of 11 male soccer players measured by the sensory evaluation test (static balance) [56]. Differently, one research did not report significant improvements in an experimental male soccer group on balance performance in the dominant leg (balance error scoring system) after eight weeks of core training [57]. Even if the training period was different, our results agree with Imai et al. [24]. Since previous authors underlined the efficiency of an 8-week integrated training on neuromuscular control and balance, this period represented a sufficient stimulus [35]. To the best of our knowledge, no other studies have analyzed the effect of specific circuit training with core exercises in the soccer off-season period.

Although the heterogeneity of studies evaluating the balance performance after core training in players makes it difficult to obtain a statement regarding soccer, it seems clear that at least four weeks of core training performed twice per week could induce benefits in athletes of several sports [20]. More evidence is needed to establish whether core exercises should be included in soccer season training or during the pre/off-season period.

Many limitations are presented in this study: (a) the sample size was small, so the statistical power resulted low for some comparisons; (b) no specific treatment was planned for the control group and participants were free to perform recreational activities; (c) no training assessment was provided during CT execution, so the intensity and fatigue levels were not evaluated.

## 5. Conclusions

The current literature highlights that training with core exercises could induce several benefits in fitness and sports. Our protocol was effective in improving the strength, core endurance, and balance of adult amateur soccer players.

Despite study limitations, our positive results showed that circuit training with core exercises appears to be a good strategy for improving the performance of adult soccer players during the off-season period. To provide more evidence, it is important to continue

investigating this kind of exercise program's effects and to also apply the intervention to the pre and in-season periods.

**Author Contributions:** Conceptualization, G.B. and M.M.; methodology, G.B., M.M. and S.M.; software, S.T. and P.M.L.; validation, M.M. and G.B.; formal analysis, M.M.; investigation, G.B.; data curation, S.T. and M.M.; writing—original draft preparation, G.B. and M.M.; writing—review and editing, S.M.; supervision, S.T. and P.M.L. All authors have read and agreed to the published version of the manuscript.

**Funding:** This research received no external funding

**Institutional Review Board Statement:** The study was conducted according to the guidelines of the Declaration of Helsinki and approved by the Bioethics Committee (prot. N 25027).

**Informed Consent Statement:** Informed consent was obtained from all subjects involved in the study.

**Data Availability Statement:** Data may be requested from authors.

**Conflicts of Interest:** The authors declare no conflict of interest.

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
