# Peer review of "Effects of Eight-Week Circuit Training with Core Exercises on Performance in Adult Male Soccer Players"

_ejihpe, doi:10.3390/ejihpe12090086_

Round 1
Reviewer 1 Report
MATERIAL AND METHODS
- In the "subjects and study design" section, the authors must clarify why the participants are amateur players (Define amateur criteria).
- Was the team made up of only 19 players?
- Why was the randomization of the participants not performed?
- Include participant selection flowchart
- Breathing was controlled during the exercises?
Author Response
Dear reviewer,
Thank you for your comments. We appreciated them and improved our manuscript.
R.:
-In the "subjects and study design" section, the authors must clarify why the participants are amateur players (Define amateur criteria).
- Was the team made up of only 19 players?
A.: Thank you, we specify it in lines 74-76.
R.:
- Why was the randomization of the participants not performed?
- Include participant selection flowchart
A.: Thank you, we report them in lines 79-83. Also, the participant’s flowchart is added in lines 91-92.
R.:
- Breathing was controlled during the exercises?
A.: Thank you, we add it in lines 140-141.
Reviewer 2 Report
Determining the benefits that core training can bring has been the subject of many studies. Unfortunately, there are still many different theories in this regard, which makes the opinions of sports theorists and practitioners inconsistent.
In this study, the authors decided to analyze the effects of a specific circuit training protocol performed during off season time, on trunk, upper and lower body strength, dynamic balance, and speed in competitive amateur soccer players.
The introduction is written correctly, it is based on the latest literature related to the subject of the work. The material and test methods are also described in great detail.
Doubts arise about the "Statistical analysis" and concern the issue of the lack of use of the two-way analysis of variance. If the authors included two groups in the study and made measurements before and after the start of circuit training, it seems necessary to use such an analysis. The results section lacks information on the differences between the groups during the pre-experiment measurement period. Table 1 shows the differences between the groups before the experiment (Pre), so it becomes apparent whether the groups were homogeneous and whether these differences were not statistically significant. Therefore, I propose to rebuild the statistical analysis, supplementing it with a two-way analysis of variance (group x measurement). Such analysis will allow for greater control of the results and for presenting more precise conclusions. In my opinion, using the difference from within group differences is a poor idea.
Author Response
Dear reviewer,
Thank you for your comment. We added the pre-experimental comparisons in table 1 as you suggested. We appreciate your second suggestion, but the two-way ANOVA lacks information without post hoc evaluation such as Tukey or Bonferroni test (with student’s t). The way you suggested is correct, but it needs the post hoc evaluation to understand where the difference is. It is the fastest method to infer, but it is not precise without the post hoc test. Differently, the paired Student t-test requests to compare each variable between two different times (pre and post), and the student’s t-test allows comparing two different variables or one variable over two groups (i.e.: SLJ pre over groups, and ∆SLJ (pre-post) over groups). In addition, the t-test is more robust than the Snedecor-Fisher F test, because while it assumes normally distributed variables, it is still a valid test for comparing approximately normally distributed groups.
It is important to remember that the pre-experiment comparison exhibits differences that could be influenced by several external factors, which can result in confounding. To avoid carrying around the beginning variability in the post-experiment, we decided to compare the differences between the post minus pre in the experimental group (∆EG), and the post minus pre in the control group (∆CG). However, if you perform the two-way ANOVA with the interaction effect and then you assess the post hoc evaluation, you still obtain similar results, but they could be distorted by their distributions. To simplify reading and interpreting results, we preferred to report the student’s t value without the F test value.

Round 2
Reviewer 2 Report
The authors well addressed all my comments and suggestions. The current version of the manuscript reaches high standards and is suitable for publication.